# The challenge of intracellular antibiotic accumulation, a function of fluoroquinolone influx versus bacterial efflux

Julia Vergalli[1,8], Alessio Atzori[2,8], Jelena Pajovic[3,5], Estelle Dumont[1,6], Giuliano Malloci [2], Muriel Masi[1,7], Attilio Vittorio Vargiu [2], Mathias Winterhalter [4], Matthieu Réfrégiers [3], Paolo Ruggerone [2] & Jean-Marie Pagès [1,8 ✉]

With the spreading of antibiotic resistance, the translocation of antibiotics through bacterial envelopes is crucial for their antibacterial activity. In Gram-negative bacteria, the interplay between membrane permeability and drug efflux pumps must be investigated as a whole. Here, we quantified the intracellular accumulation of a series of fluoroquinolones in population and in individual cells of *Escherichia coli* according to the expression of the AcrB efflux transporter. Computational results supported the accumulation levels measured experimentally and highlighted how fluoroquinolones side chains interact with specific residues of the distal pocket of the AcrB tight monomer during recognition and binding steps.

[1] UMR_MD1, U-1261, Aix-Marseille University, INSERM, IRBA, MCT, Marseille, France. [2] Department of Physics, University of Cagliari, 09042 Monserrato, CA, Italy. [3] DISCO beamline, Synchrotron Soleil, Saint-Aubin, France. [4] Department of Life Sciences and Chemistry, Jacobs University Bremen, 28719 Bremen, Germany. [5] Present address: University of Belgrade, Faculty of Physics, 11001 Belgrade, Serbia. [6] Present address: Institut National Supérieur d'Agronomie et de Biotechnologies (INSAB), Université des Sciences et Techniques de Masuku (USTM), BP941 Franceville, Gabon. [7] Present address: Institut de Biologie Intégrative de la Cellule (I2BC), Université Paris Saclay, CNRS, CEA, 91198 Gif-sur-Yvette, France. [8] These authors contributed equally: Julia Vergalli, Alessio Atzori, Jean-Marie Pagès ✉email: Jean-Marie.PAGES@univ-amu.fr

Antibiotic discovery and development are urgently needed to face the deadly danger of the emergence and spread of multi-drug-resistant bacteria[1–3]. This concern is especially true for Gram-negative pathogens belonging to the ESKAPE group, e.g., *Pseudomonas aeruginosa*, *Acinetobacter baumannii*, and carbapenem-resistant Enterobacteriaceae[4,5]. In Gram-negative bacteria, the envelope permeability has been studied for a long time. However, due to its complex structure, composition, and diversity of membrane transporters, there is still a serious gap in our understanding of antibiotic translocation into living bacteria[6–9]. Several pharmaceutical companies have carried out high-throughput screening campaigns and reported a large number of attractive antibacterial molecules. Unfortunately, most of them have not entered into the market, mostly because of weak intrabacterial accumulation levels leading to toxicity issues at effective doses[8–12].

Fluoroquinolones (FQs) represent one of the most commonly used antibiotics to treat human and veterinary infections due to their excellent bioavailability, good tissue penetration, relatively low toxicity, and broad-spectrum activity[13–15]. Numerous FQ derivatives have been synthesized by modifying the chemical structure of the 1,8 naphthyridine core in order to increase their antibacterial efficiency and improve their pharmacokinetics[16–20]. These modifications have led to the synthesis of norfloxacin and ciprofloxacin, two antibiotics with increased activity against Gram-negative bacteria, which are still marketed[13].

Quinolone resistance is mostly due to mutational alteration of the drug targets that can be exacerbated concomitantly by a decreased outer-membrane permeability and an increased active efflux[8,12,16,21]. These membrane-associated mechanisms of resistance constitute the first line of defense and contribute to lower the intracellular concentration of drugs at the vicinity of their targets[6–8,21,22]. Numerous studies have been conducted on FQs for improving pharmacokinetics and reducing cytotoxicity, or for making molecules able to overcome bacterial resistances such as hybrid FQ conjugates[23–26]. However, the ability of such compounds to reach effective cytoplasmic concentrations remains a key factor for their antimicrobial activity, a question that has not been properly addressed until now. The Innovative Medicine Initiative programs (www.imi.europa.eu) have supported the development of appropriate methods and concepts to measure translocation and to quantify the intrabacterial concentration of drugs[9,27]. Using fluorimetric assays, we previously showed that the molecular properties of three FQs dictate their capacity to accumulate in *E. coli* expressing or not the major multidrug efflux transporter AcrB, in complex with the adapter protein AcrA (in the following denoted as AcrAB)[8,27]. The SICAR (Structure Intracellular Concentration Activity Relationship) index has been proposed for quantifying accumulation of FQs and other antibiotics[8,10,12,21,22].

This study aims at getting insights into the molecular bases of drug translocation by comparing a large set of FQs in *E. coli* as a model of Gram-negative bacteria, as well as their sensitivity to AcrB-mediated transport. Drug susceptibility assays, spectro- and micro-fluorimetry, molecular docking, molecular dynamics (MD) simulations, and binding free-energy calculations were combined to investigate antibacterial drug activity, accumulation, and extrusion, respectively, in isogenic strains expressing different levels of AcrAB. Our findings provide a robust correlation between internal drug concentration and efflux activity, which paves the way for the design of predictive FQ accumulation rules in Gram-negative bacteria.

## Results

**Antibacterial activity**. To determine how active efflux contributes to the antibacterial activity and intracellular accumulation of quinolone antibiotics, we selected nalidixic acid (NAL), flumequine (FLU), and 20 commercially available FQs. These compounds differ in their molecular and physicochemical properties while they share a similar core structure (Table 1). Susceptibility assays were performed using isogenic *E. coli* K12 strains expressing various levels of AcrAB: AG100 (wild type), AG100A (AcrAB-efficient mutant), and AG102 (over-expressing AcrAB)[27] (see "Methods").

Minimal inhibitory concentrations (MICs) of quinolones and doses for early killing (DEK) were measured (see Methods). Overall, MICs and DEK were inversely proportional to the level of AcrAB expression: MICs were increased by 2–64-fold in *E. coli* WT (AG100) and by 8–128-fold in the efflux-proficient mutant (AG102) compared with the efflux-deficient mutant (AG100A) (Supplementary Table 1).

To quantify the activity of antibiotics in the efflux-deficient mutant, as well as the impact of efflux on these activities, MICs and DEK measured in AG100A were plotted against the ratio of activities measured in AG102 (Fig. 1a, b). Only PEF and BAL displayed a different behavior with a 4-fold increase for their DEK compared with their MIC. DEK of BAL was also more affected by efflux activation in AG102.

NAL exhibited the lowest activity on AG100A. Although slightly affected by efflux, NAL remained the less active quinolone on AG102 followed by FLU. Some FQs such as TRO or PAZ showed an intermediate level of activity on AG100A, but were modestly affected by efflux activation. In contrast, NAD and LEV displayed potent activity on AG100A, but were also hypersusceptible to efflux: the MIC of NAD and LEV was potentiated by 128- and 64-fold, upon efflux activation, respectively. SPA, GEM, and TOS were the most active FQs on AG100A, and were found to be susceptible to efflux. Other compounds, such as PEF, BAL, LOM, and NOR, showing intermediate activities on AG100A, also presented an intermediate level of activity on AG102. As a whole, FLE, MOX, OFL, ENR, CIP, and GAT also displayed similar activities and efflux susceptibilities.

Compounds were grouped by colors according to their generation in Fig. 1a, b. First-generation FQs are characterized by a weak activity along with a low-efflux susceptibility. For the second and third generations, increased activity was directly associated with efflux susceptibility, except for PAZ. Fourth-generation FQs are the most potent with a moderate-to-low susceptibility to efflux. Interestingly, all FQs with a C7-aminopyrrolidine substituent instead of a C7 piperazine (except PAZ, which has an aminocyclopropyl) aggregated in the lower-left part of the graph (TOS, CLI, GEM, and TRO), corresponding to compounds with high activity and low-efflux sensitivity.

The determination of the DEK on AG100A for each tested compound allowed for their ranking according to their penetration/activity efficacy (Fig. 1b):

SPA ≫ NAD, GAT, LEV, TOS, CLI, GEM > CIP, ENR, PEF, PAZ, BAL > FLE, LOM, OFL, MOX, TRO, NOR > ENO > FLU ≫ > NAL.

Some molecules are clearly different, e.g., SPA or NAL, while some others form clusters: "CIP, ENR, PEF, PAZ, BAL" or " FLE, LOM, OFL, MOX, TRO, NOR".

This analysis was completed by the measurement of their DEK in AG102, and we observed the following ranking of efflux susceptibility (Fig. 1b):

TRO < CLI, GEM, CIP, ENR, PEF, PAZ, FLE, LOM, OFL, MOX, ENO, NAL, < GAT, LEV, TOZ, NOR, FLU < SPA, BAL < NAD.

In summary, the tested FQs distribute in two large clusters with some marked exceptions.

**FQ accumulation in efflux minus background**. Spectrofluorimetry and microspectrofluorimetry were used to measure the FQ concentrations inside the bacterial population (Fig. 2a, b) and the single

**Table 1 Tested quinolones and their physicochemical characteristics.**

| | Used abbreviation | Molecular weight | Molecular volume | Artificial membrane permeability | ChromlogD | ClogD7.4 | XLOGP3 | Globularity | Rot. bonds | IC50 (µg ml⁻¹) | | Fluorescent compounds used in fluorimetry assays |
|---|---|---|---|---|---|---|---|---|---|---|---|---|
| | | | | | | | | | | S. typhimurium | E. coli | |
| Nalidixic acid | NAL | 232.2 | 241.5 | 51 | 1.31 | −0.3 | 1.4 | 0.07 | 2 | 65.1 | 50 | |
| Flumequin | FLU | 261.2 | | | | | 2.9 | | 1 | | 4 | |
| Ciprofloxacin | CIP | 331.3 | 311.5 | 9 | 0.09 | −1.4 | −1.1 | 0.07 | 3 | 0.25 | 0.3 | Yes |
| Enrofloxacin | ENR | 359.4 | 353.5 | 82 | 1.89 | | −0.2 | | 4 | 0.57 | | Yes |
| Enoxacin | ENO | 320.3 | 304.5 | 12 | 0.02 | −1.7 | −0.2 | 0.08 | 3 | | | |
| Fleroxacin | FLE | 369.3 | 339.5 | 70 | 1.7 | 0.1 | −0.1 | 0.06 | 4 | | | Yes |
| Lomefloxacin | LOM | 351.3 | 336 | 28 | 0.67 | −1.1 | −0.8 | 0.15 | 3 | | | Yes |
| Norfloxacin | NOR | 319.3 | 311.5 | 8 | 0.08 | −1.5 | −1 | 0.07 | 3 | | 0.9 | Yes |
| Nadifloxacin | NAD | 360.4 | 346.5 | 60 | 1.48 | 0.6 | 2.9 | 0.09 | 2 | | | |
| Ofloxacin | OFL | 361.4 | 339.5 | 42 | 0.94 | | −0.4 | | 2 | 0.69 | 0.35 | Yes |
| Pefloxacin | PEF | 333.4 | 332.5 | 56 | 1.23 | 0.2 | 0.3 | 0.06 | 3 | | 1.2 | Yes |
| Balofloxacin | BAL | 389.4 | 381.5 | 66 | 0.66 | | 0.6 | | 5 | | | |
| Levofloxacin | LEV | 361.4 | 339.5 | 42 | 0.94 | −0.3 | −0.4 | 0.07 | 2 | 0.43 | 0.29 | |
| Pazufloxacin | PAZ | 318.3 | 283.5 | 32 | 0.53 | | −0.8 | | 2 | | | Yes |
| Sparfloxacin | SPA | 392.4 | 371 | 159 | 0.91 | −0.7 | 0.1 | 0.12 | 3 | 0.42 | 0.2 | |
| Tosufloxacin | TOS | 404.3 | 360.5 | 38 | 2.02 | | 0.4 | | 3 | | | |
| Clinafloxacin | CLI | 365.8 | 329 | 59 | 0.49 | −2.1 | 0.4 | 0.18 | 3 | | | |
| Gemifloxacin | GEM | 389.4 | 367.5 | 32 | −0.05 | −2.6 | −0.7 | 0.48 | 5 | | | |
| Moxifloxacin | MOX | 401.4 | 381.5 | 138 | 1.38 | −1.7 | 0.6 | 0.15 | 4 | 0.48 | 1.6 | |
| Gatifloxacin | GAT | 375.4 | 360.5 | 45 | 0.3 | −1.2 | −0.7 | 0.15 | 4 | 0.3 | | Yes |
| Trovafloxacin | TRO | 416.4 | 353.5 | 214 | 2.13 | −1.6 | 0.3 | 0.2 | 3 | | | |

*Artificial membrane permeability* permeability of a substance from a donor compartment through an artificial phospholipid membrane into an acceptor compartment, *ChromlogD* chromatographic hydrophobicity[55], *ClogD7.4* logarithm of theoretical n-octanol/water partition coefficient corrected for ionization at pH 7.4, *XlogP3* logP (logarithm of n-octanol/water partition coefficient for neutral forms) determined with an additive model[56], *Globularity* three-dimensionality of compounds from ref. [38], *Rot. bonds* rotatable bonds, number of flexible single bonds, *Fluorescent compounds* fluorescence signals were also detected for balofloxacin, levofloxacin, and moxifloxacin, but these compounds were not used in the fluorimetric assays mentioned in this study. IC50 from refs. [57,58], and unpublished results are underlined.

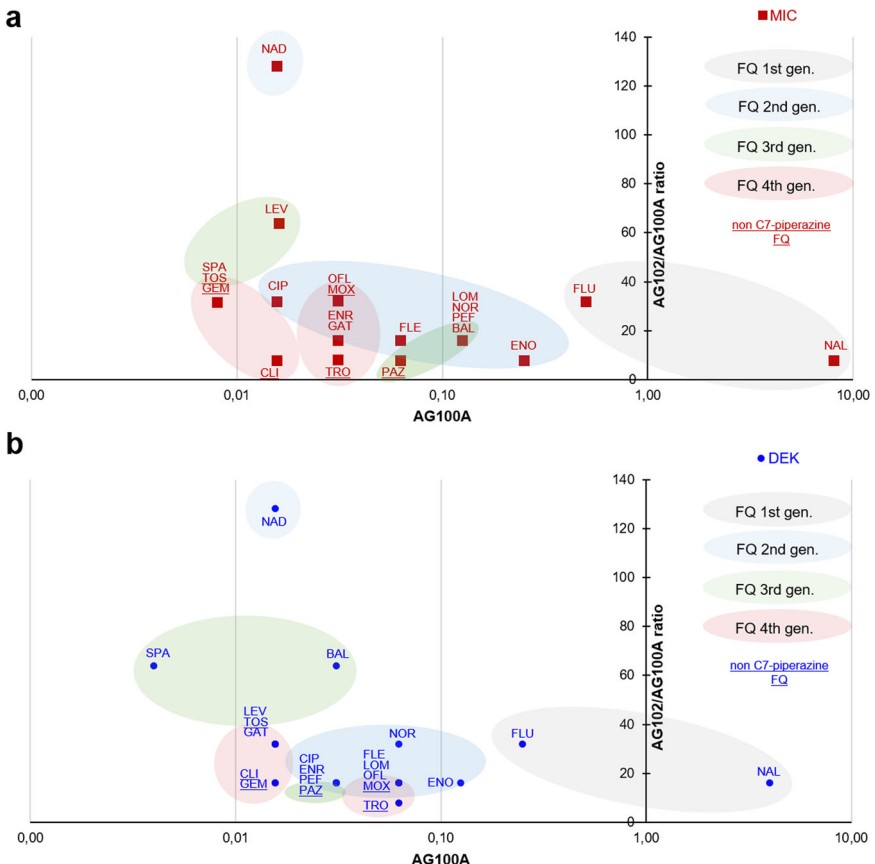

**Fig. 1 Quinolone activities measured with MIC and DEK vary according to their capacity of membrane translocation and sensibility to efflux.** Quinolone activities measured on the efflux-deficient mutant (AG100A) were plotted on the x axis (influence of the capacity of membrane translocation on activities) against the ratio of activities measured on the efflux-proficient compared with the -deficient mutant (AG102/AG100A) plotted on the y axis (influence of the AcrAB efflux pump sensibility on activities). **a** Red squares, MIC values. **b** Blue circles, DEK (dose necessary for early killing) values. See also Supplementary Table 1. To optimize the readability of the figure, the abscissa is represented as a logarithmic scale, and the quinolone names are abbreviated as mentioned in Table 1.

bacterial cell (Fig. 2c, d), respectively[28]. Only FQs yielding a strong usable fluorescent signal intensity to quantify their intracellular concentration were considered: CIP, ENR, FLE, LOM, NOR, OFL, PEF, PAZ, and GAT (Table 1). Assays were performed with various FQ concentrations and different incubation times. As previously reported for FLE, the accumulation plateau was obtained at 5 min of incubation[28].

$SICAR_{IN.100A}$ corresponds to the maximal intracellular concentration of a given compound in the efflux-deficient mutant AG100A, thus indicating its influx capacity by membrane translocation. Figure 3 shows differences between $SICAR_{IN.100A}$ for the tested FQs (P value = 0.00000727). GAT exhibited the highest $SICAR_{IN.100A}$ with ~34,000 molecules accumulated per bacteria in 5 min. CIP, ENR, NOR, and PEF showed similar $SICAR_{IN.100A}$ with a decrease of about 15% compared with GAT. Accumulation of OFL corresponded to an intermediate level between the "CIP, ENR, NOR, PEF" group and the "PAZ, FLE, LOM" group. This latter cluster of compounds had the lowest $SICAR_{IN.100A}$ among all tested FQs, with 13,700 molecules/bacterial cell/5 min for LOM.

Interestingly, for CIP, ENR, NOR, and PEF with no $R_8$ substituent (see Fig. 3 for a graphical representation of all $R_i$s), a similar $SICAR_{IN.100A}$ was observed. GAT, which exhibited the highest $SICAR_{IN.100A}$, has a $R_8$ O-methyl moiety. OFL and PAZ, characterized by an average and low level of accumulation, respectively, contained a cyclization of their radicals $R_8$ and $R_1$.

FLE and LOM have a $R_8$ fluorine and displayed the lowest $SICAR_{IN.100A}$ among our tested compounds.

The three FQs exhibiting a noticeable $SICAR_{IN.100A}$ have a $R_1$ cyclopropyl (GAT, CIP, and ENR). However, the effect of $R_1$ on $SICAR_{IN.100A}$ seemed to be secondary compared with the $R_8$ substituents; e.g., NOR, PEF, and LOM with a methyl in $R_1$ accumulate at high (NOR and PEF) or low levels (LOM).

All tested FQs contain a piperazine in $R_7$, except PAZ, which features an aminocyclopropyl group. The localization of methyl-substituting piperazine did not seem to affect the $SICAR_{IN.100A}$ of FQs. GAT and LOM, which were the highest and the lowest accumulated FQs, shared the same 3-methyl piperazine. Similarly, PEF and FLE accumulated at high and low levels, despite the same N-methyl piperazine.

Concerning the effect of concentration on $SICAR_{IN.100A}$, no saturation effect on intracellular accumulation was observed for CIP, NOR, and LOM with increasing concentrations up to 12.5 μM (Supplementary Fig. 1). On the contrary, $SICAR_{IN.100A}$ of ENR, PEF, FLE, PAZ, and OFL slightly saturated when the concentration increased from 5 to 12.5 μM compared with the $SICAR_{IN.100A}$ measured at the lowest concentrations (from 1 to 5 μM).

**AcrAB expression levels and FQ accumulation.** Accumulation of the various FQs has also been measured in AG100 and AG102. To further monitor the efflux impact, $SICAR_{EFF.100}$ and

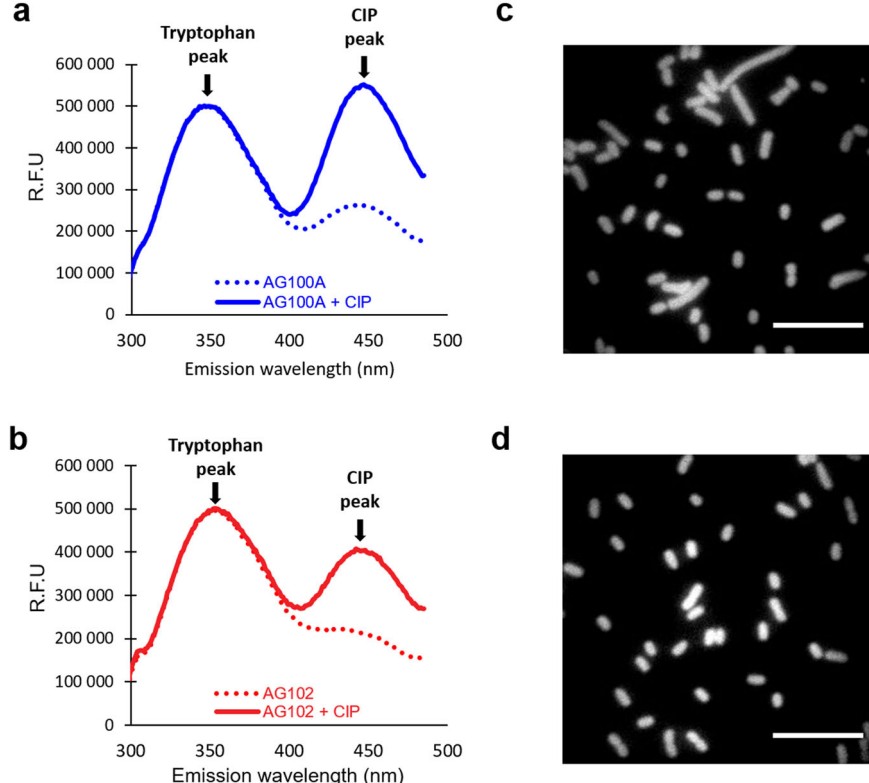

**Fig. 2 UV fluorescence tracks FQs in bacteria.** Spectrofluorimetry spectra and microspectrofluorimetry imaging of CIP accumulation measured in AG100A and AG102 populations (**a** and **b**, respectively) and in living individual cells (**c** and **d**). Bacterial suspensions were incubated for 5 min without and with CIP at 5 μM, and suspensions were sampled from the same incubation mixture for spectrofluorimetry and microspectrofluorimetry measurements: for spectrofluorimetry analyses, bacterial pellets were lysed, and emission spectra of bacteria incubated without (dotted lines) or with (solid lines) CIP were measured with a spectrofluorimeter (**a** for AG100A and **b** for AG102). FQ concentrations in lysates were then calculated according to calibration curves (see "Methods" and ref. [27]). For microspectrofluorimetry measurements in living individual cells, bacterial pellets were resuspended in buffer and analyzed by deep–ultraviolet fluorescence imaging (examples are shown in **c** for AG102 and in **d** for AG100A, scale bars indicate 10 μm). Images were then analyzed and corrected to compare the various FQs and bacterial strains (see "Methods" and ref. [27]) as obtained in Supplementary Fig. 3.

$SICAR_{EFF.102}$ were calculated as the ratio of the accumulated concentration in AG100A to the accumulated concentration in AG100 and AG102, respectively[27]. Figure 4a shows the $SICAR_{EFF.100}$. GAT and ENR displayed the highest $SICAR_{EFF.100}$, with ratios of 3 and 3.6, respectively. LOM also appeared efflux-sensitive with a $SICAR_{EFF.100}$ of 2.4. Importantly, the distribution was preserved with these three FQs in the $SICAR_{EFF.102}$ ranking (Fig. 4b).

From Fig. 4a, NOR, CIP, and OFL had a similar $SICAR_{EFF.100}$, around 1.5, which is close to the lowest index of PAZ (1.1). The ranking of the NOR–CIP–OFL trio was conserved in $SICAR_{EFF.102}$, whereas PAZ seemed to be more affected by AcrAB overexpression and shifted to a medium location (Fig. 4b). Independently of the $SICAR_{EFF}$ measured, FLE remained at an intermediate position, and its ratio slightly increased with AcrAB expression in bacteria.

Overall, the ranking of $SICAR_{EFF.102}$ (Fig. 4b) was quite similar to the $SICAR_{EFF.100}$, except for PAZ and PEF. For PAZ, we may hypothesize that the increase in AcrAB expression can change the accumulation level since its internal concentration is relatively moderated (Fig. 3). In the case of PEF, the ratio remained fairly close (2.0 vs. 1.5) and may be associated with its influx efficacy.

Interestingly, the effect of a concentration of 12.5 μM on $SICAR_{EFF.102}$ was inversely correlated with the $SICAR_{EFF.102}$ measured at 5 μM for most tested FQs (Supplementary Fig. 2). PEF, CIP, and OFL, which had the lowest $SICAR_{EFF.102}$ at 5 μM, had a $SICAR_{EFF.102}$ increased at 12.5 μM, while the $SICAR_{EFF.102}$ of

PAZ, LOM, and ENR, which were the highest at 5 μM, was decreased at 12.5 μM. $SICAR_{EFF.102}$ of NOR and FLE, which was at an intermediate position at 5 μM, remained constant at 12.5 μM.

**Effect of efflux blockade on accumulation of individual cells**. Microspectrofluorimetry measurements were performed on individual cells of AG102 without and with carbonyl cyanide m-chlorophenylhydrazine (CCCP), which is used herein as an experimental protonophore, to study three FQs in the same strain in the presence and in the absence of active efflux (Supplementary Fig. 3). ENR was highly susceptible to efflux compared with NOR and FLE. The accumulation of ENR in individual cells is greatly increased in the presence of CCCP compared with FLE and NOR, and shows the highest variability of accumulated concentrations in the cells.

Interestingly, GAT, which showed a noticeable permeation efficacy, also appears very efflux- sensitive. About LOM, its intrabacterial concentration is strongly affected by two permeability barriers, e.g., low outer-membrane permeation and active efflux.

**Molecular modeling**. A systematic ensemble-docking campaign of the FQs of interest was performed with a major focus on the two main putative binding sites of AcrB known from structural studies[29,30], namely the access pocket of the loose monomer ($AP_L$) and the distal pocket of the tight monomer ($DP_T$). According to our results, the majority of docking poses were

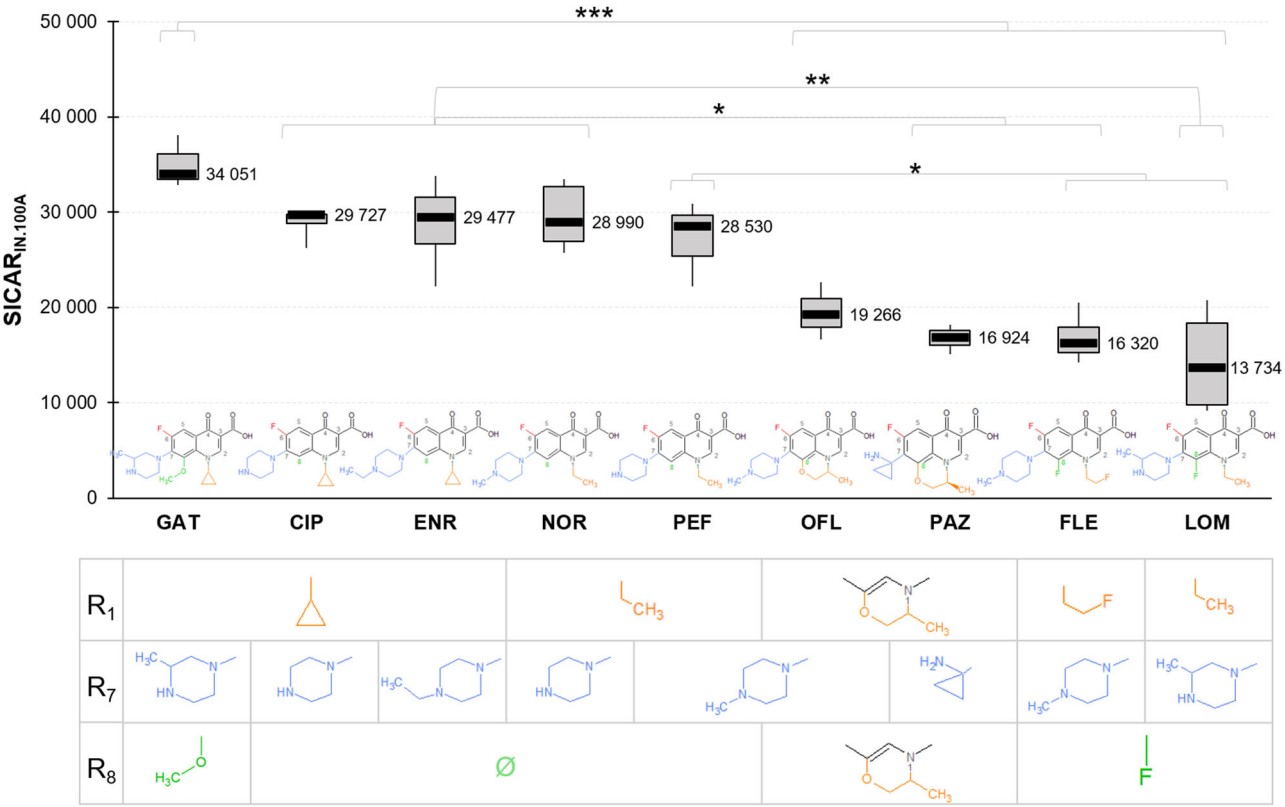

**Fig. 3 Influx: FQs have different capacities of membrane translocation measured with SICAR$_{IN.100A}$.** SICAR$_{IN.100A}$ was obtained from the accumulated concentrations (molecules/cell) in the efflux-deficient mutant AG100A for 5-min incubation with 5 μM of FQs. Data are represented by a box-and-whisker plot (the ends of the box are the upper and lower quartiles, the median is marked by a black line inside the box, and the whiskers are the two lines outside the box that extend to the highest and lowest measurements). Substituents $R_1$, $R_7$, and $R_8$ of the FQ structures are indicated in the table under each corresponding FQ. ANOVA and Tukey's post hoc tests were performed to determine differences between FQs ($n = 32$ biologically independent samples, $\omega^2 = 0.69$, degree of freedom = 8). ***$P < 0.001$; **$P < 0.01$; *$P < 0.05$. Data normality was checked by the Shapiro–Wilk test, and homogeneity of variance by the Fligner–Killeen test.

localized in the DP$_T$, while a minimal number were found in the AP$_L$. DP$_T$ is supposed to be visited during extrusion by all the substrates captured by the efflux system[31], and it is also believed to be the putative recognition site of low-molecular-mass AcrB substrates[29]. Therefore, in the following, we focus only on this site (see Supplementary Table 2 for a list of residues lining the DP$_T$).

Due to the intrinsic limitations of the docking scoring function employed, the ranked average binding affinities predicted by Autodock VINA failed to replicate the trend observed by spectrofluorimetry quantification of drug accumulation (reported in Fig. 4a). However, the analysis of the docking poses provided multiple relevant configurations used as good starting points to perform two MD simulations for each FQ–AcrB complex. In particular, we employed a reduced model of AcrB not containing the transmembrane domain whose reliability has been thoroughly assessed in previous studies[32,33]. For each simulation considered, a cluster analysis was carried out on the MD trajectory followed by a binding free-energy evaluation through the Molecular Mechanics/Generalized Born Surface Area method. The results, shown in Fig. 5 and collected in Supplementary Table 3, reproduce fairly well the experimental trend of Fig. 4a, as confirmed by a correlation coefficient of ~0.82 (Supplementary Fig. 4). From Fig. 5, it is possible to rank the considered FQs in three classes according to their average binding affinity to the DP$_T$: low affinity (PAZ, NOR, and CIP), medium affinity (LOM, OFL, GAT, FLE, and PEF), and high affinity (ENR).

**Dynamic behavior of NOR, FLE, and ENR**. To further investigate the molecular rationale behind the difference in binding affinity measured, eight additional MD simulations were performed for a representative compound from each class, namely NOR (low affinity), FLE (medium affinity), and ENR (high affinity). The starting conformations of the supplementary MD simulations were taken from docking poses selected to have a broader sampling of the DP$_T$[32].

The majority of the selected poses for NOR were found inside the hydrophobic trap (HP trap, lined by F136, F178, F610, F615, and F628), which is a critical recognition site reported for AcrB substrates and efflux inhibitors[34]. Nonetheless, all of them left this region during the simulations reaching different portions of the DP$_T$. While exploring this region, NOR showed generally low-binding affinities (Supplementary Table 4 and Supplementary Fig. 5), confirming the trend extracted from the first two simulations (Supplementary Table 3 vs. Supplementary Table 4). Despite these low affinities, hydrophobic contacts between NOR and the DP$_T$ residues seem to play a more important role than hydrogen bonds (Fig. 6).

On the other hand, trajectories of FLE and ENR were considerably more stable compared with NOR, remaining close to their starting position, partly inside the HP trap. Overall, the interaction profiles mirrored the behavior of the low-, medium-, and high-affinity representative compounds. Opposite to NOR, both FLE and ENR showed a much more comparable interaction profile, characterized by multiple and durable hydrophobic

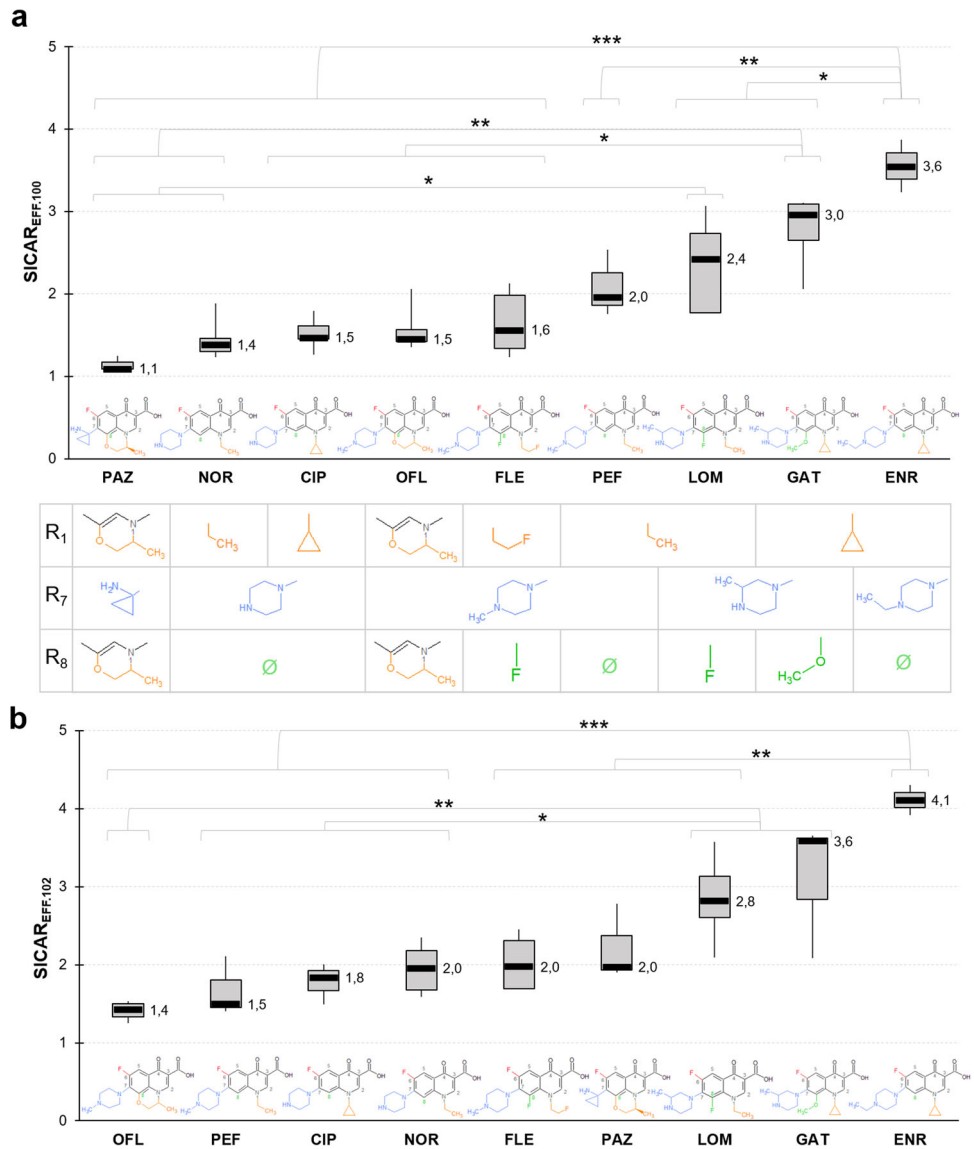

**Fig. 4 Efflux: FQs have different susceptibilities to efflux measured with SICAR$_{EFF.100}$ and SICAR$_{EFF.102}$.** SICAR$_{EFF.100}$ **a** and SICAR$_{EFF.102}$ **b** were obtained by the ratio of the accumulated concentrations in the efflux-deficient mutant AG100A to the accumulated concentrations in the wild-type strain AG100 (**a**) or in the efflux-proficient mutant AG102 (**b**), for 5-min incubation with 5 μM of FQs. See also Supplementary Fig. 3. Data are represented by a box-and-whisker plot. Substituents R$_1$, R$_7$, and R$_8$ of the FQ structures are indicated in the table under each corresponding FQ of panel **a** ANOVA and Tukey's post hoc tests were performed to determine differences between FQs (**a** $n = 35$ biologically independent samples, $\omega^2 = 0.69$, degree of freedom = 8; **b** $n = 31$ biologically independent samples, $\omega^2 = 0.70$, degree of freedom = 8). ***$P < 0.001$; **$P < 0.01$; *$P < 0.05$. Data normality was checked by the Shapiro–Wilk test and homogeneity of variance by the Fligner–Killeen test.

interactions, and by hydrogen bonds with residue R620 (Fig. 6). The stabilizing role of R620 together with F628 is confirmed by their contribution to the free energy of binding-reported ΔG′ (Supplementary Fig. 6). Moreover, a consistently higher number of common residues contributing to ΔG' can be observed only in ENR and FLE trajectories, while the quinolone core allows for interactions between all of the three FQs and F178, F615, and Y327 (Supplementary Fig. 6). Note that the contacts observed between FLE/ENR and the residues of the HP trap (and nearby hydrophobic residues) are present also in the available experimental structures of wild-type AcrB in complex with its substrates doxorubicin, minocycline, rhodamine-6G, and puromycin[29–36].

Interestingly, the analysis of water molecules surrounding the compounds within the first hydration shell (Supplementary Table 4) unveils that (i) NOR is surrounded by a higher number of water molecules than the other two FQs, regardless of the region occupied inside the DP$_T$; (ii) FLE and ENR trajectories present a clearly distinct number of water molecules depending on the region occupied: lower inside the bottom region of the DP$_T$ and higher outside of it.

The comparison of ΔG′ for the selected FQs (Supplementary Table 4, and Supplementary Figs. 5 and 6) reveals how (i) the average affinities predicted for the three FQs by enhancing the sampling of the DP$_T$ maintain the trend shown previously in Fig. 5 (ENR > FLE > NOR); (ii) ENR possesses on average larger negative binding affinities when found inside the bottom region of DP$_T$ rather than outside of it; (iii) ΔG′ values for FLE appear to be comparable, within error bars, regardless of the position occupied inside the DP$_T$; (iv) NOR showed a considerably low likeness for the HP trap (starting docking poses left during the simulations), but higher affinity for the bottom region than the other portions of the DP$_T$.

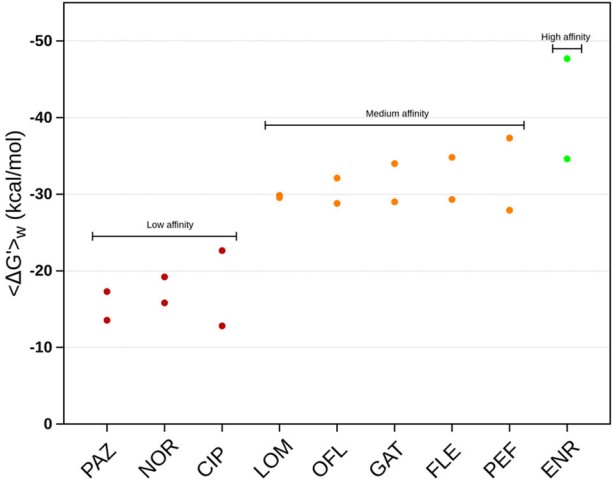

**Fig. 5 Calculations of the free energy of binding to the DPT of AcrB allow to distinguish three classes of FQs.** For each compound, we reported the two weighted average free energies of binding obtained from the clusters populated by at least 10% of the total conformations sampled during the two performed MD simulations and computed using the Molecular Mechanics/Generalized Born Surface Area method (see Methods for details and Supplementary Table 3 for the considered values). All values are expressed in units of kilocalories per mole (kcal mol$^{-1}$).

## Discussion

In this study, we used *E. coli* strains with different efflux activity to identify trends in the antibacterial activity of FQs and the intracellular concentrations in correlation with their sensitivity to efflux. MIC determination and resazurin-based killing assay exhibit different key parameters, such as the incubation time in the presence of the drug (18 h for MIC assays vs. 3–5 h for killing assays), as well as the ratio of drug concentration/cell concentration (100 times higher in the case of killing assay). Thus, the resazurin-based killing assay describes the early bacterial response to drug exposure with a higher inoculum size that likely corresponds to that of infection sites[37].

Determination of SICAR$_{IN}$ indicates efficient translocation of GAT, CIP, ENR, and NOR compared with PAZ and FLE. According to Richter et al., accumulated drug concentrations are likely related to the flexibility (number of rotatable bonds) and shape (globularity) of the drug[38]. The considered FQs share low globularity and low flexibility that are supposed to provide fast translocation and high accumulation. However, their accumulation levels differ remarkably, suggesting that those features alone do not explain the observed variations. FQ generations 2, 3, and 4, exhibit similar structures that differ only in the substituents at R$_1$, R$_7$, and R$_8$ of the quinolone core. R$_1$, R$_7$, and R$_8$ substituents appear the key factors for penetration and target binding for which new compounds are developed[39,40]. N1 cyclopropyl and R$_8$ unsubstituted seemed to be related to high SICAR$_{IN}$. These results corroborate the important occurrence of N1 cyclopropyl as R$_1$ substituent. Furthermore, bulky groups at C$_8$ position have been considered to affect the overall molecular steric conformation[41], consistent with the low SICAR$_{IN}$ of OFL and PAZ whose third cycle in R$_8$ may induce a steric hindrance for their translocation into the cells. All the tested FQs have a C7-piperazine group except PAZ. This moiety has been shown to increase the drugs' ability to pass the bacterial membrane resulting in enhanced activity[40]. Piperazine substitutions in studied FQs do not alter SICAR$_{IN}$, although it has been reported that numerous changes in piperazine moiety can affect the activity of the drug[13].

FQs exhibit different sensitivity to AcrAB expression levels, e.g., the substantial decrease for NAD activity compared with the reduced ones for TRO. Activities have decreased up to 128-fold in the strain overexpressing AcrAB, highlighting the major impact of efflux on the FQ activity. ENR and PAZ illustrate the effect of efflux: high SICAR$_{IN}$ of ENR is balanced by its high sensitivity to efflux, while low SICAR$_{IN}$ of PAZ goes with a weak sensitivity to efflux, which balances the respective DEK. The level of AcrAB expression can also change the internal concentration of FQs and reflect the affinity of specific substituents to the putative binding pocket DP$_T$ of AcrB.

The robustness of our computational protocol is confirmed by the agreement between the trend of the calculated free energies of binding and that extracted from SICAR$_{EFF}$. Importantly, ENR, FLE, and NOR revealed different interaction patterns, thus explaining the difference of their AcrB sensitivity. Specifically, ENR and FLE have multiple and resilient hydrophobic interactions with the HP trap, while NOR shows a considerably lower affinity for this region. The hydration of the compounds is also different, being NOR surrounded by more water molecules than ENR and FLE. Moreover, NOR hydration is substantially constant in all DP$_T$ regions visited by the molecule, while that of ENR and FLE depends on their positions inside the DP$_T$. A better hydration may result in a more efficient screening of the interaction with DP$_T$ residues. Thus, the preference of NOR for limiting durable interactions, especially with residues of the HP trap, could not be optimal for triggering allosteric conformational changes needed to AcrB to accomplish its function.

When looking at the combination of the efflux and influx indexes (SICAR$_{IN}$, SICAR$_{EFF}$), FQs generally tend to present only one favorable index among the two. CIP seems to be the only FQ that combines the two favorable indexes. Interestingly, this molecule could be used as a platform to modulate the side chains in order to preserve a high SICAR$_{IN}$ and decrease the SICAR$_{EFF}$. This demonstrates how challenging is the design of compounds able to overcome the membrane barrier, and how necessary are studies focused on the permeability problem as a whole. The future generation of quinolones should be powerful and shows a molecular profile ensuring the activity against multi-drug-resistant bacteria; thus, predictive molecular designs allowing the internal accumulation versus efflux pump efficacy will represent a key aspect. Our study paves the way for rational pharmacomodulation improving the design of FQs that bypass the membrane-associated mechanisms of resistance.

## Methods

**Bacteria and media**. All *E. coli* strains used in this study, AG100 (wild type), AG100A (*acrB*::Kan derivative, efflux-deficient mutant), and AG102 (over-expressing AcrAB due to a gain-of-function *mar* mutation, efflux-proficient mutant) have been previously described[27].

**Chemicals**. All the 21 quinolones used in this study were purchased from Sigma-Aldrich. Nine compounds were used in fluorescence studies. The physicochemical parameters of the studied chemicals are listed in Table 1.

**Drug susceptibility assay**. MIC values of antibiotics were determined by the microdilution method in liquid Mueller Hinton II medium by using twofold standard microdilutions, in microplates according to the guidelines of the Clinical & Laboratory Standards Institute (CLSI, http://clsi.org/). MICs were determined in the absence and in the presence of the efflux pump inhibitor phenylalanine arginine β-naphthylamide (PAβN, Sigma) at a final concentration of 20 μg ml$^{-1}$. MIC values were read after 18 h of incubation at 37 °C. Experiments were carried out at least in triplicate, and the resulting medians were presented.

**Resazurin-based bacterial viability assay**. The resazurin-based bacterial viability assay or killing assay was used to evaluate the metabolic activity of cells[8], and to determine the dose of FQ needed for early killing (DEK). When resazurin (non-fluorescent blue-colored compound) penetrates into bacterial cells, it is reduced by metabolically active bacterial cells to resorufin (fluorescent, pink-colored product), which results in a color change that can be determined with measurement of the fluorescent signal of resorufin (Exc. 568 nm, Em. 660 nm). Briefly, bacteria grown at

**a**

| | | Hydrophobic contacts | | | | | | | | | | | | | | | | | | | | H-bond | | | | |
|---|---|---|---|---|---|---|---|---|---|---|---|---|---|---|---|---|---|---|---|---|---|---|---|---|---|---|---|
| | F136 | V139 | L177 | F178 | I277 | A279 | P326 | Y327 | V571 | M573 | F610 | V612 | F615 | F617 | I626 | F628 | L668 | P669 | A670 | V672 | M862 | S134 | S180 | N274 | R620 | S630 |
| NOR | 20 | 0 | 19 | 26 | 27 | 0 | 0 | 18 | 0 | 0 | 0 | 0 | 31 | 22 | 0 | 11 | 18 | 17 | 14 | 15 | 11 | 0 | 7 | 0 | 0 | 0 |
| FLE | 53 | 33 | 0 | 44 | 20 | 0 | 10 | 19 | 18 | 15 | 36 | 25 | 26 | 0 | 19 | 48 | 12 | 0 | 16 | 19 | 0 | 0 | 0 | 0 | 8 | 5 |
| ENR | 48 | 36 | 0 | 55 | 35 | 15 | 0 | 17 | 0 | 24 | 33 | 0 | 35 | 18 | 30 | 48 | 13 | 0 | 0 | 24 | 0 | 6 | 8 | 6 | 10 | 0 |

**b**

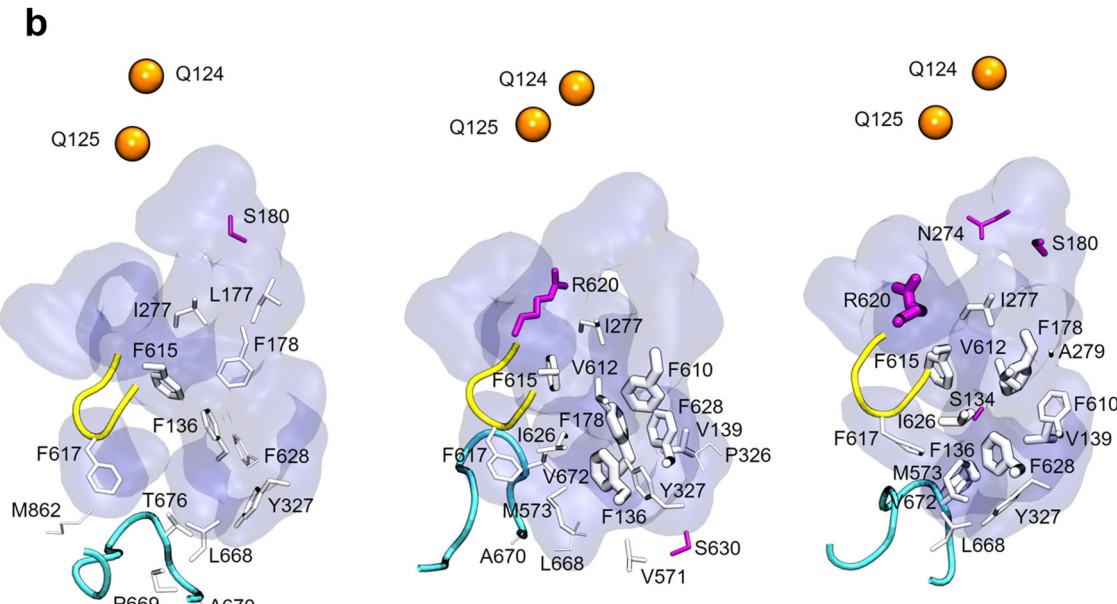

**Fig. 6 NOR, FLE, and ENR have different interaction profiles with AcrB DPT residues. a** Table reported the total occurrence of hydrophobic contacts and H bonds between NOR, FLE, and ENR and the residues lining the $DP_T$. The contacts were extracted for each compound by merging the ten MD simulations in a single trajectory, and an error of ~6% should be considered. **b** Representative view of the $DP_T$ from NOR, FLE, and ENR simulations, where the residues involved in interactions with the selected FQs are represented as sticks (white for hydrophobic contacts and magenta for H bonds). Stick width is proportional to the frequency of the considered interaction. Protein residues lining the $DP_T$ are displayed as a blue surface. The switch- and the bottom loop are represented, respectively, as a yellow and cyan tube. Exit gate residues are shown as orange spheres. For the list of residues defining $DP_T$, switch-, bottom loop, and exit gate see Supplementary Table 2 and ref. [45].

37 °C in Mueller Hinton broth to mid-exponential phase (corresponding to 0.6 optical density units at 600 nm ($OD_{600}$) were diluted in Mueller Hinton broth to obtain a bacterial suspension of around $2.10^7$ cells $ml^{-1}$. In 96-well clear-bottom black microplates, 170 µl of the bacterial suspension was mixed with 20 µl of the viability resazurin dye and 10 µl of individual FQs at various concentrations. The same bacterial suspensions incubated without quinolones, were used as controls. The fluorescent signal of resorufin was monitored during 5 h at 37 °C with the plate reader (Tecan, Infinite 200 PRO). The drug-killing percent, corresponding to the difference of fluorescence intensities measured in the control and the studied conditions, was determined at the first time point of the fluorescence plateau measured in the control condition (plateau corresponding to 100% fluorescence and thus 100% viability/0% killing).

DEK for "Dose for Early Killing" was determined from the killing assay; it is the lowest concentration (dose) of drug inducing bacterial killing.

**FQ accumulation in intact bacteria**. Accumulation assays were performed as previously described, in triplicate, and repeated at least three independent times[27,28]. Bacteria were grown at 37 °C in LB (Luria-Bertani broth) to mid-exponential phase. The bacterial suspension was centrifuged at $6000 \times g$ for 15 min at 20 °C and concentrated tenfold by resuspension of the pellets in 1/10 of the initial volume in 50 mM sodium phosphate buffer at pH 7 supplemented with $MgCl_2$ (5 mM) (NaPi–MgCl$_2$ buffer) to obtain a density of $6 \times 10^9$ CFU (colony-forming units) per ml. In glass culture tubes, 2.4 ml of the bacterial suspension was

incubated for 5 min at 37 °C with FQs at 1, 5, or 12.5 µM (final volume 3 ml), in the absence or in the presence of the efflux blocker CCCP (carbonyl cyanide m-chlorophenylhydrazine) used at 10 µM that collapses the energy-driven force needed by the efflux pump[27]. Bacterial suspensions incubated without antibiotics, with or without CCCP, were used as controls. Suspensions (800 µl for spectrofluorimetric measurements or 400 µl for microspectrofluorimetric measurements) were then loaded on NaPi–MgCl$_2$ buffer (1100 or 550 µl, respectively) and centrifuged at $9000 \times g$ for 5 min at 4 °C to collect the washed bacteria.

CFUs were determined to control the viability of cells, and no change in viability was observed during the experimental time. It must be noted that the ratio cell/antibiotic concentration were different in the MIC and in the accumulation assay carried out using starving conditions during a limited incubation time (5–30 min).

**Spectrofluorimetry**. After centrifugation, pellets corresponding to 800 µl of bacterial suspensions were lysed with 500 µl of 0.1 M Glycin-HCl pH 3 overnight at room temperature. After a centrifugation for 15 min at $9000 \times g$ at 4 °C, 400 µl of lysates were mixed with 600 µl of Glycin-HCl buffer, and emission spectra were measured with a spectrofluorimeter (Fluoromax 4 spectrofluorimeter (Horiba-Jobin Yvon). Excitation/emission range wavelengths (nm) used for detection of FQ fluorescence signal with spectrofluorimetry were as the following: 275/435–450, 275/435–450, 290/485–510, 290/500–515, 290/435–450, 320/420–435, 275/435–450, 290/440–450, and 275/435–445, for CIP, ENR, GAT, OFL, FLE, PAZ, NOR, LOM, and PEF, respectively. The fluorescence signals were corrected by the tryptophan peak of the bacteria

(fluorescence emission between 330 and 360 nm) to obtain a fluorescence signal per bacterial cell[42]. Calibration curves were carried out to determine the quantity of molecules accumulated per cell: various concentrations of the studied FQ were mixed with bacterial lysates at $OD_{600}$ of 4.8 and measured using a spectrofluorimeter ($n = 3$ replicates)[27].

**DUV microspectrofluorimetry**. To detect the FQ fluorescence from single-bacteria background, pellets corresponding to 400 µl of bacterial suspensions were resuspended in 40 µl of NaPi–MgCl₂ buffer. About 0.5 µl of resuspended pellets were deposited between two quartz coverslips and analyzed by deep–ultraviolet (DUV) fluorescence imaging at DISCO Beamline (Synchrotron SOLEIL)[27]. Bacterial cells were first located in brightfield before excitation in DUV under a microscope (Zeiss Axio Observer Z-1). Emission was collected through a Zeiss ultrafluar objective at 100× with glycerin immersion. The FQ fluorescence were recorded after excitation using a dichroic mirror at 300 nm (OMEGA Optical, Inc., USA) through appropriate emission band-pass filters (OMEGA Optical, Inc., USA; SEMROCK, USA)[27]. Excitation/emission filters (nm) were as the following: 275/ 420–480, 275/412–438, and 290/420–480, or 435–455, 290/435–455, 290/420–480, and 275/420–480, or 420–460, 275/420–480, or 420–460, or 402–447, 275/402–447, for CIP, ENR, GAT, OFL, FLE, NOR, LOM, and PEF, respectively. For the tryptophan fluorescence, the emission was passing through an emission band-pass filter at 327–353 nm (SEMROCK). Fluorescence images were recorded by a back-illuminated ultraviolet (BUV) electron-multiplying charge-coupled device (EM CCD) (Princeton PIXIS 1024 BUV). The whole setup (microscope, stages, filters, and camera) was controlled by Micro-Manager[27].

The images were analyzed with ImageJ (Rasband, W.S., ImageJ, U.S. National Institutes of Health, Bethesda, Maryland, USA, http://imagej.nih.gov/ ij/). The illumination heterogeneities were corrected before background subtraction. First, threshold was automatically adjusted using a triangle algorithm; thereafter, bacteria were analyzed as the remaining particles. The mean intensity coming from each bacterium was automatically calculated considering its pixel area. The fluorescence intensities for each FQ were normalized first by the intensities in tryptophan filter collected immediately after the drug fluorescence signal was acquired. The obtained ratios were subsequently normalized by the mean value of the ratios corresponding to the control samples for each condition. Each bacterium from one image was averaged and considered as one emitter. For each condition, five different localizations with a minimum of ten bacteria per field of view were recorded and averaged. The experiment was independently repeated at least three times.

**Molecular docking**. Molecular docking was performed with the program Auto-dock VINA[43]. The ten most populated molecular conformations sampled from MD trajectories of each FQ in a solvated water box (www.dsf.unica.it/translocation/ db)[44] and ten available X-ray structures of AcrB have been considered to account for ligand and receptor flexibility (Supplementary Table 2). Thus, for each FQ, a total of 100 independent runs were performed, using a docking volume of 125 × 125 × 110 Å³ covering the whole protein. The first five ranked poses for each run were considered, resulting in 500 docking poses for each compound. We focused on the two main putative binding sites of AcrB known from structural studies[29,30], namely the access pocket of the loose monomer ($AP_L$) and the distal pocket of the tight monomer ($DP_T$). Criteria adopted for the selection of the poses were: (i) the total number of docking poses in contact with at least 40% of the residues lining the two pockets[32,45], (ii) the average values of the corresponding "binding affinities" (evaluated through the docking scoring function), and (iii) the frequency of contacts established between the compound and the residues lining the pockets.

**MD simulations and binding free-energy calculations**. Two docking poses of the AcrB–FQ complex located inside the $DP_T$ were selected and used as starting points for 100-ns-long MD simulations for each compound, performed using Amber18[46]. Due to the substantial number of compounds to be screened, a reduced model of the AcrB protein not containing the transmembrane domain has been employed. The quality of the simplified model adopted has been validated in previous applications[32,33]. Each selected system was immersed in a box containing TIP3P water molecules[47] and an adequate number of K⁺ counterions, in order to neutralize the negative net charge of the system. An osmolarity of 0.15 M was reached by adding an appropriate number of K⁺/Cl⁻. The ff14SB version of the all-atom Amber force field[48] was adopted for AcrB, while for the FQs, we considered the general Amber force field parameters taken from www.dsf.unica.it/translocation/db[44]. In order to guarantee a slow equilibration phase while keeping the asymmetric structure of the reduced system in accordance with the crystallographic data, the equilibration and the production runs were performed as follows: to rearrange the position of waters and ions, structural relaxation was performed in the presence of soft restraints (1 kcal mol⁻¹ Å⁻²) on all the non-hydrogen atoms of the protein and the ligand. In the second and third steps, the restraints were kept only on backbone and Cα atoms, respectively, and on the non-hydrogen atoms of ligand. Finally, restraints were removed from the ligand and from a selection of residues having at least one atom within 8 Å from the ligand. In all steps, the structure of the solute from the previous step was used as a target for restraints, and up to 10,000 optimization steps were

performed using the conjugate-gradient algorithm. Next, annealing up to 340 K was performed in 2 ns, using the same setup as in the last step of the relaxation described at the previous point, and constant volume and temperature conditions (NVT ensemble). This was followed by quenching to 310 K in 3 ns, and then a 1-ns-long equilibration with the same setup as above, but in the NPT ensemble. Finally, a productive run of 100 ns was performed by applying partial restraints to the system, namely to all heavy atoms of the protein, but those having at least one atom within 8 Å from the ligand. The last conformation from previous dynamics was used as a target for structural restraints. The trajectories were saved every 100 ps, resulting in 1000 conformations for each system. A time step of 2 fs was used during all these steps of the equilibration protocol. MD simulations for each system were carried out using the PMEMD module of Amber14[49] with a time step of 4 fs in NVT ensemble, after application of the hydrogen mass repartitioning[50]. A Langevin thermostat using a collision frequency of 1 ps⁻¹ and a Berendsen isotropic barostat[51] maintained a constant temperature, and an average pressure of 1 Atm, respectively. Long-range electrostatic interactions were calculated using the particle mesh Ewald method with a cutoff of 9 Å[52].

Hydrogen bonds (cut-off distance of 3.5 Å and angle of 120° between donor and acceptor atoms) and hydrophobic contacts (distance of <2.5 Å between the centers of mass of the molecule and of the residues of the binding region) were analyzed using a customized tcl script for VMD[53]. Only the H bonds and hydrophobic contacts established, respectively, for more than 5% and 10% of the total simulation time, were considered relevant. A cluster analysis using the *cpptraj* module of AmberTools14 identified the most populated states sampled during the simulation of each system with a fixed clustering radius of 3.5 Å. For each simulation, only the cluster populated for more than 10% of the total simulation time (Supplementary Tables 3 and 4) was further taken into account for evaluating the free energy of binding of the molecule defined as

$$\Delta G_b = \Delta E_{MM} + \Delta G_{solv} - T\Delta S_{conf}, \qquad (1)$$

where $\Delta E_{MM}$ and $\Delta G_{solv}$ are the differences in the molecular mechanics energy and in the solvation-free energy, respectively. $\Delta G_b$ is calculated using the Molecular Mechanics/Generalized Born Surface Area method[54].

**Statistics and reproducibility**. The data plotted in Fig. 1 (reported in Supplementary Table 1) are the medians of independent experiments.

The data plotted in Figs. 2, 3, 4, 5, 6 and in Supplementary Fig. 5 are collected in the excel file data.xls included in Supplementary Data 1.

The results of $SICAR_{IN.100A}$, $SICAR_{EFF.100}$, and $SICAR_{EFF.102}$ plotted in Figs. 3 and 4 were obtained from at least $n = 3$ independent experiments. Statistical analysis was performed using the computing environment R (R Development Core Team, 2005). ANOVAs (degrees of freedom = 8) with Tukey's post hoc tests were used to determine differences between SICAR indexes shown in Figs. 3 and 4. $P$ values between 0.01 and 0.5 were considered significant (*), $P$ values between 0.001 and 0.01 were considered very significant (**), and $P$ values < 0.001 were considered extremely significant (***). Size effects ($\omega^2$) are reported in the corresponding figure legends. The data normality and homoscedasticity were checked by the respective Shapiro–Wilk and Fligner–Killeen tests.

In Fig. 5 (Sheet Delta G′—all in Supplementary Data 1 and data collected in Supplementary Table 3), we reported the mean values of the free energies of binding extracted from two MD simulations for each compound. The statistical analysis was performed by considering the clusters populated for more than 10% of the simulation time, and mean values were calculated by weighting the values according to the cluster population. Thus, the number of data considered for the statistical analysis of each simulation is different, ranging from 1 to 3.

The frequencies of hydrophobic contacts and H bonds reported in Fig. 6 (Sheet Contacts in Supplementary Data 1) were evaluated by merging the ten MD simulations in a single trajectory for each considered compound (NOR, FLE, and ENR). The same ten MD simulations were considered to evaluate the free energies of binding and the associated standard deviations reported in Supplementary Fig. 5 (Sheet Delta G′—selected in Supplementary Data 1 and data reported in Supplementary Table 4). As for the analysis yielding the data in Fig. 5, for each compound, weighted averages and standard deviations were estimated according to the populations of the single clusters for each simulation, ranging in the number of considered clusters from 1 to 3. These values were then considered to evaluate the free energies of binding and the associated standard deviations reported in Supplementary Fig. 5.

**Reporting summary**. Further information on research design is available in the Nature Research Reporting Summary linked to this article.

## Data availability

All data supporting this study are available within the article and its Supplementary Information or are available from the corresponding author on reasonable request.

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

## Acknowledgements

We thank R. A. Stavenger, A. Davin-Regli, and J. M. Bolla for their fruitful discussions; B. Pineau and V. Rouam for their assistance during the assays. The research leading to these results was conducted as part of the TRANSLOCATION consortium, and has received

support from the Innovative Medicines Initiatives Joint Undertaking under Grant Agreement n°115525, resources that are composed of financial contribution from the European Union's seventh framework program (FP7/2007–2013) and EFPIA companies in kind contribution. A.A., G.M., A.V.V. and P.R. also received support from the National Institutes of Allergy and Infectious Diseases Project number AI136799. This work was also supported by Aix-Marseille University and Service de Santé des Armées, and by Soleil program (projects #20160883, #99170096, #20171060, and #20180683).

## Author contributions

J.V., A.A., J.P., E.D., A.V.V., G.M., and M.M.: investigation and data curation. M.M., M.W., P.R., and J-M.P.: resources. M.W., M.R., P.R., and J-M.P.: supervision. All the coauthors: writing, review, and final editing.

## Competing interests

The authors declare no competing interests.
