## [Peer Review File · Communications Biology]

Reviewers' comments:

Reviewer #1 (Remarks to the Author):

The manuscript represents an interesting contribution in the field of antibiotic accumulation inside bacterial cells and the antibacterial activity and effectivity as well. The focus was put on Gram-negative bacteria and fluoroquinolones and on the study of correlation between molecular characteristics, structural properties and antibacterial effect. The authors try to summarize and analyze a lot of obtained data in the manuscript.

I have a few comments/questions/proposals how to improve the manuscript:

1/ Authors use spectrofluorimetry, microspectrofluorimetry and fluorescence imaging techniques, but they do not demonstrate any fluorescence spectrum or fluorescence image. This should be done and incorporated into the main text.

2/ Figure S1 is only a scheme, it does not contain a specific result. The reference to it in line 167 is confusing and not comprehensible.

3/ Figure S2 - The meaning of this image is not clear. What exactly represents the x-axis? What is its physical meaning?

4/ Several abbreviations are used without explanation (e.g. CCCP, DUV), there are some missing subscripts (e.g. MgCl₂).

Reviewer #2 (Remarks to the Author):

This manuscript reports the quantification intracellular accumulation of fluoroquinolones in *E. coli* cells expressing different levels of the AcrB multidrug efflux pumps. The authors also predict the binding affinities of a variety of fluoroquinolones using AutoDock Vina. In addition, they perform MD simulation, utilizing Amber, to elucidate how AcrB exports these antibiotics. The work is nicely done and should be acceptable for publication in *Commun. Biol.* There are a couple of very minor comments that the authors may want to consider before publication.

Minor comments:

In the MD simulation section of the text (p. 9-10), the authors should indicate that they just used the periplasmic domain (not the full-length protein structure) for the calculation.

Fig. 5b: Please label all of the residues.

Fig. 5b: Why is that some of the sticks representing the side chains in the figure are thicker than others? Please provide with a reason in the figure legend.

Fig. 5b: Please indicate the G-loop is colored yellow in the figure legend or label the G-loop in the figure.

Fig. 5b: Please indicate the cyan loop, which should be the flexible F-loop at the proximal site, in the figure legend or label it in the figure.

Fig. 5b: Why are the orange balls corresponding to the antibiotics were drawn outside the pump? Does it mean that they are already exported? Please indicate it in the figure legend.

Manuscript COMMSBIO-19-1829-T

Reviewers' comments and corresponding responses indicated by >>

Reviewer #1 (Remarks to the Author):

The manuscript represents an interesting contribution in the field of antibiotic accumulation inside bacterial cells and the antibacterial activity and effectivity as well. The focus was put on Gram-negative bacteria and fluoroquinolones and on the study of correlation between molecular characteristics, structural properties and antibacterial effect. The authors try to summarize and analyze a lot of obtained data in the manuscript.

I have a few comments/questions/proposals how to improve the manuscript:

1/ Authors use spectrofluorimetry, microspectrofluorimetry and fluorescence imaging techniques, but they do not demonstrate any fluorescence spectrum or fluorescence image. This should be done and incorporated into the main text.

>> As suggested by the reviewer, we have added a new figure (Figure 2) to illustrate the spectrofluorimetry and microspectrofluorimetry methods performed in this manuscript with the example of fluorescence spectra and fluorescence images recorded during CIP accumulation for two strains, AG100A and AG102.

2/ Figure S1 is only a scheme, it does not contain a specific result. The reference to it in line 167 is confusing and not comprehensible.

>> This figure has been deleted for clarity

3/ Figure S2 - The meaning of this image is not clear. What exactly represents the x-axis? What is its physical meaning?

>> These points have been detailed in the Figure S2 legend.

4/ Several abbreviations are used without explanation (e.g. CCCP, DUV), there are some missing subscripts (e.g. MgCl₂).

>> Definitions of abbreviations are included in the revised version.

Reviewer #2 (Remarks to the Author):

This manuscript reports the quantification intracellular accumulation of fluoroquinolones in *E. coli* cells expressing different levels of the AcrB multidrug efflux pumps. The authors also predict the binding affinities of a variety of fluoroquinolones using AutoDock Vina. In addition, they perform MD simulation, utilizing Amber, to elucidate how AcrB exports these antibiotics. The work is nicely done and should be acceptable for publication in *Commun. Biol.* There are a couple of very minor comments that the authors may want to consider before publication.

Minor comments:

In the MD simulation section of the text (p. 9-10), the authors should indicate that they just used the periplasmic domain (not the full-length protein structure) for the calculation.

>> As suggested by the Reviewer we added a sentence specifying the use of a simplified model of AcrB so that the whole paragraph at p.9-10 now reads:
However, the analysis of the docking poses provided multiple relevant configurations used as good starting points to perform two MD simulations for each FQ-AcrB complex. In particular, we employed a reduced model of AcrB not containing the transmembrane domain whose reliability has been thoroughly assessed in previous studies^{32,46}. For each simulation considered, a cluster analysis was carried out on ...

Fig. 5b: Please label all of the residues.

>> Done in the revised version (Note that Fig. 5b is now Fig. 6b.)

Fig. 5b: Why is that some of the sticks representing the side chains in the figure are thicker than others? Please provide with a reason in the figure legend.

>> We specified in the Figure legend that the width of each stick is proportional to the frequency of the considered interaction.

Fig. 5b: Please indicate the G-loop is colored yellow in the figure legend or label the G-loop in the figure.

>> We indicated explicitly in the Figure legend that the G-loop is colored yellow.

Fig. 5b: Please indicate the cyan loop, which should be the flexible F-loop at the proximal site, in the figure legend or label it in the figure.

>> We indicated explicitly in the Figure legend that the F-loop is colored cyan.

Fig. 5b: Why are the orange balls corresponding to the antibiotics were drawn outside the pump? Does it mean that they are already exported? Please indicate it in the figure legend.

>> The orange balls correspond to the Exit Gate residues as specified in the Figure legend.

We changed Figure 5b (now figure 6b) in the revised version of the manuscript and modified the corresponding legend according to the answers provided above. The new legend now reads :

Figure 6. NOR, FLE, and ENR have different interaction profiles with AcrB DP_T residues. (a) Total occurrence of hydrophobic contacts and H-bonds between NOR, FLE, ENR and the residues lining the DP_T shown respectively as red, orange, and green bars. Labels of residues involved in H-bonds are in bold. (b) Representative view of the DP_T from NOR, FLE and ENR simulations, where the residues involved in interactions with the selected FQs are represented as sticks (white for hydrophobic contacts and magenta for H-bonds). Sticks width is proportional to the frequency of the considered interaction. Protein residues lining the DP_T are displayed as a blue surface. The switch-loop and the bottom-loop are represented respectively as a yellow and cyan tube. Exit gate residues are shown as orange spheres. For the list of residues defining DP_T, switch-loop, bottom-loop, and exit gate see Supplementary Table 2 and Ref.

REVIEWERS' COMMENTS:

Reviewer #1 (Remarks to the Author):

The authors nicely completed the manuscript and figures. Regarding figures, it is still necessary to add the ticks in Figure 2 and the axes with ticks in Figure 5 and 6.